# Revisiting Microbial Diversity in Hypersaline Microbial Mats from Guerrero Negro for a Better Understanding of Methanogenic Archaeal Communities

**DOI:** 10.3390/microorganisms11030812

**Published:** 2023-03-22

**Authors:** José Q. García-Maldonado, Hever Latisnere-Barragán, Alejandra Escobar-Zepeda, Santiago Cadena, Patricia J. Ramírez-Arenas, Ricardo Vázquez-Juárez, Maurilia Rojas-Contreras, Alejandro López-Cortés

**Affiliations:** 1Centro de Investigación y de Estudios Avanzados del Instituto Politécnico Nacional, Unidad Mérida, Mérida 97310, Yucatán, Mexico; 2Centro de Investigaciones Biológicas del Noroeste (CIBNOR), La Paz 23205, Baja California Sur, Mexico; 3Microbiome Informatics Team, EMBL-EBI, Hinxton CB10 1SD, UK; 4Centro de Investigaciones Químicas, Universidad Autónoma del Estado de Morelos, Cuernavaca 62209, Morelos, Mexico; 5Departamento de Agronomía, Universidad Autónoma de Baja California Sur, La Paz 23080, Baja California Sur, Mexico

**Keywords:** bacteria, methanogens, mcrA, hypersaline microbial mats, metagenomics

## Abstract

Knowledge regarding the diversity of methanogenic archaeal communities in hypersaline environments is limited because of the lack of efficient cultivation efforts as well as their low abundance and metabolic activities. In this study, we explored the microbial communities in hypersaline microbial mats. Bioinformatic analyses showed significant differences among the archaeal community structures for each studied site. Taxonomic assignment based on 16S rRNA and methyl coenzyme-M reductase (*mcrA)* gene sequences, as well as metagenomic analysis, corroborated the presence of Methanosarcinales. Furthermore, this study also provided evidence for the presence of Methanobacteriales, Methanomicrobiales, Methanomassiliicoccales, *Candidatus* Methanofastidiosales, Methanocellales, Methanococcales and Methanopyrales, although some of these were found in extremely low relative abundances. Several *mcrA* environmental sequences were significantly different from those previously reported and did not match with any known methanogenic archaea, suggesting the presence of specific environmental clusters of methanogenic archaea in Guerrero Negro. Based on functional inference and the detection of specific genes in the metagenome, we hypothesised that all four methanogenic pathways were able to occur in these environments. This study allowed the detection of extremely low-abundance methanogenic archaea, which were highly diverse and with unknown physiology, evidencing the presence of all methanogenic metabolic pathways rather than the sheer existence of exclusively methylotrophic methanogenic archaea in hypersaline environments.

## 1. Introduction

Knowledge of archaeal diversity has significantly grown in recent years with the development of bioinformatic methodologies and the unceasing generation of high-throughput sequencing data and cultivation [1]. The expansion and reshaping of the archaeal phylogenetic tree changed the ecological and evolutionary importance of this domain. Archaea have been considered a major fraction of microbial diversity [2,3,4], living in exceedingly diverse habitats, including the most environmentally extreme [4]. They are metabolically diverse, including mesophiles and (hyper-)thermophiles, anaerobes and aerobes, autotrophs and heterotrophs, with a large diversity of putative archaeal symbionts as well as previously unknown acetogens and different groups of methanogens [3].

Previous studies on methanogens in hypersaline environments reported that the diversity of methanogenic archaea consisted largely of methylotrophic methanogens [5,6,7]. However, recent studies also showed evidence of hydrogenotrophic methanogenesis in hypersaline microbial mats [8,9], suggesting that the archaeal diversity in Guerrero Negro could be underestimated probably due to their low abundances under these environmental conditions. Previous assays of enriched samples from Exportadora de Sal, S.A. (ESSA), sites ESSA-A1 (6% salinity) and ESSA-A9 (17% salinity) with trimethylamine (TMA), suggested the presence *Methanomassiliicoccus* and Thermoplasmatales, related to the class Thermoplasmata, in these ecosystems [10]. These methanogenic groups were considered to represent the seventh order of methanogenic archaea [11]. Members of the proposed order Methanoplasmatales were previously reported within different environments, including marine habitats, soil, the intestinal tracts of termites and mammals [2,12] and, more recently, in smooth hypersaline microbial mats from Shark Bay [9].

Physiological, genomic and metatranscriptomic studies of Methanomassiliicoccales revealed that this group employed a methylotrophic methanogenesis pathway by reducing methanol, methylamines and, presumably, also methylated sulphides to methane [13,14]. In addition, genomic analyses revealed that Methanomassiliicoccales lacked the methyl branch of the Wood–Ljungdahl pathway (e.g., *Methanomicrococcus blatticola* and *Methanosphaera* spp.), which employs external hydrogen as an electron donor, thus extending the spectrum of organisms exhibiting hydrogen-dependent methylotrophic methanogenesis [15].

Although the occurrence of hydrogen-dependent methylotrophs was also linked to the presence of methylated compounds, these cannot dismutate the substrate and strictly depend on the donation of electrons from hydrogen and/or formate to reduce the methyl group to methane [16]. To date, this metabolic capacity is present in members of the recently discovered class Methanonatronarchaeia, members of the order Methanomassiliicoccales and *Candidatus* Methanofastidiosales [17,18], members of the phyla Ca. Bathyarchaeota [19] and Ca. Verstraetearchaeota [20], as well as in the species *Methanoplasma termitum* in the order Methanomassiliicoccales [17] and *M. blatticola* in the order Methanosarcinales [21].

Despite recent advances in phylogenetics and research on methanogenic pathways in different environments, the key methanogenic players, as well as the contribution of different substrates to methane formation, remain elusive. In this study, we performed a deep exploration of highly diverse methanogenic archaeal communities in the hypersaline environments in Guerrero Negro, Baja California Sur, Mexico, through high-throughput amplicon sequencing of 16S rRNA and *mcrA* genes, as well as shotgun metagenome sequencing approaches, to expand the knowledge of this rare biosphere and evaluate the metabolic potential of methanogenic pathways in these ecosystems.

## 2. Materials and Methods

### 2.1. Fieldwork

Samples of soft-smooth laminated microbial mats (cores of 10 mm width × 5 mm depth) were collected in triplicate in June 2019 from four brine concentrator ponds (Area 1, Area 4 near Area 1, Area 4 near Area 5 and Area 5), located at Exportadora de Sal, S.A., in Guerrero Negro, Baja California Sur, Mexico (Appendix A; Table 1). All samples were stored in 2 mL cryogenic vials (Nalgene; Thermo Fisher, Carlsbad, CA, USA) containing 1.5 mL RNAlater^®^ (Thermo Fisher, Carlsbad, CA, USA) and kept on ice until taken to the laboratory, where they were stored at −20 °C. Environmental variables such as salinity, temperature, dissolved oxygen and pH were estimated in situ (Table 1).

### 2.2. Library Preparation of 16S rRNA and mcrA Genes

The first batch of DNA extractions was carried out in triplicate for each ESSA area, using 0.25 g of the first 5 mm of microbial mat samples. Microbial cells were lysed with TissueLyser LT (QIAGEN, Hilden, Germany), and DNA extracted with the DNeasy PowerSoil Kit (QIAGEN, Germantown, MD, USA) following the manufacturer’s protocol. To assess potential cross-contamination from the kit reagents, a negative control (column with no sample) was processed together with the target samples. The DNA quality and quantity were visualised with a 1% agarose gel. The 16S rRNA gene amplicons were amplified in triplicate with the universal primer set 515F-Y and 926R (Appendix A) [22], covering the V4–V6 regions for bacteria and archaea. Thermocycling conditions were carried out with the Veriti 96-well Fast thermocycler (Applied Biosystems, Foster City, CA, USA) as follows: initial denaturation at 95 °C for 2 min, 25 cycles at 95 °C for 45 s, 52 °C for 45 s, 68 °C for 90 s and a final extension at 68 °C for 5 min. The PCR reactions (25 μL) included 2 μL of DNA, 0.5 μL of F/R primer (10 μM) and 12.5 μL of GoTaq master mix (Promega, Madison, WI, USA).

For the *mcrA* gene, a second batch of DNA extractions was performed with the QIAGEN “DNeasy PowerBiofilm” kit (GmbH, Hilden, Germany). Each sample was macerated with a polypropylene pestle and an electrical homogeniser (VWR, Wayne, PA, USA), and the aqueous phase containing the biomass was recovered and processed according to the kit’s instructions. The DNA integrity and concentration were assessed with standard agarose gel electrophoresis and spectrophotometric reads using a NanoDrop Lite spectrophotometer (NanoDrop Technologies, Wilmington, DE, USA). The PCR amplifications (25 µL) were conducted in triplicate as follow: 6.5 µL sterile water, 2.5 µL of each primer solution (10 µM), 12.5 µL GoTaq master mix (Promega, Madison, WI, USA) and 1 µL (10 ng µL^−1^) DNA. The gene amplifications were performed using the primers mlas-mod-F and mcrA-rev-R, reported in [23] (Appendix A). The thermocycling conditions were as follows: one cycle of 95 °C for 5 min, five touchdown cycles of 95 °C for 30 s, 60 °C for 45 s, diminishing 1 °C per cycle and an elongation step of 72 °C for 1 min, followed by 35 cycles of 95°C for 30 s, 54.5 °C for 30 s, 72 °C for 1 min and a final elongation step at 72 °C for 5 min. All PCR assays were carried out in a Thermocycler C-1000 (Bio-Rad, Berkeley, CA, USA), and the resulting amplicons were analysed using standard gel electrophoresis.

### 2.3. High-Throughput Sequencing

The 16S rRNA and *mcrA* PCR products were purified using AMPure XP magnetic beads (Beckman Coulter Genomics, Brea, CA, USA). The purified PCR amplicons were indexed with the Nextera XT Index kit (Illumina, San Diego, CA, USA) according to the Illumina 16S Metagenomic Sequencing Library Preparation handbook. Barcoded PCRs were re-purified as indicated above and subsequently quantified with a Qubit 3.0 fluorometer (Life Technologies, Petaling Jaya, Malaysia). The proper size of the libraries was verified on an QIAxcel Advanced system (GmbH, Hilden, Germany). Paired-end sequencing (2 × 250 bp) was performed at CINVESTAV Mérida with the MiSeq platform (Illumina, San Diego, CA, USA), using a 500-cycle MiSeq Reagent Nano Kit v. 2. Raw sequencing data produced in this study were deposited in NCBI under the BioProject accession number PRJNA821506.

### 2.4. Bioinformatic Analyses of 16S rRNA Metaprofiling

The obtained reads had an average length of 250 bp. The demultiplexed archives were analysed with the QIIME2 (2019.1) pipeline [24] to obtain the Amplicon Sequence Variants (ASVs), and the DADA2 package was used for denoising, error correction and removing chimeras with the “consensus” method [25,26]. Taxonomic assignment of representative ASVs sequences was performed with the V-SEARCH classifier plugin [27] against the SILVA reference database (v.138.1). Alignment of representative ASVs was filtered for non-conserved and gapped positions with the MAFFT algorithm [28] to build a phylogenetic tree with the FastTree package [29]. The ASVs tables were further analysed with the phyloseq [30], vegan [31] and ggplot2 [32] libraries and loaded into R-studio software 4.2.1 version. To evaluate beta diversity based on the abundance and phylogenetic relationship of archaeal taxa, a Principal Coordinate Analysis (PCoA) was performed to calculate the weighted UniFrac distances [33]. In addition, the alpha diversity (observed ASVs, Shannon index) was estimated from all samples [34].

### 2.5. Processing the mcrA Amplicon Metaprofiling

Quality control of raw reads was performed using FASTP v.1.14.5 [35] with the default options. High-quality trimmed sequences were used for amplicon reconstruction using Flash2 v.2.2.00 [36], setting the minimum overlapping length to 3 nt. Amplicons that were shorter than 400 bp were discarded, and chimeric sequences were detected de novo based on abundance using Usearch v.6.1.544 [37].

A database of the McrA protein sequences was built using all entries from Uniprot (UniProt Consortium 2021) for EC:2.8.4.1 (38 sequences from SwissProt and 6270 from TrEMBL, downloaded on 5 August 2021), and a BLASTX v.2.9.0+ [38] analysis of the high-quality and non-chimeric amplicons was computed, setting the parameters to report the top five matches. Sequences with no hits were discarded for the upstream analysis. The “Operational Taxonomic Units” (OTUs) were generated at 97% of nucleotide identity, and abundances were rarefied to 8299 using the rarefy function in the R vegan library v.2.4-6 [31]. Clusters of size one, represented only in one sample, were discarded from the OTUs table.

A phylogenetic tree was constructed using as reference a McrA protein alignment kindly provided by Dr. Luke McKay (Montana State University, Department of Land Resources and Environmental Sciences). We used the TAXIT v.0.9.2 command-line tool of the TAXTASTIC package [39] to generate the reference package necessary for PPlacer v. 1.1.alpha17-6-g5cecf99 [40]. Protein sequences in the OTUs representative sequences were deduced from BLASTX versus the McrA database retrieved from UniProt, and a multisequence alignment of queries and references was generated using MAFFT v.7.487 [28]. The alignment was trimmed using Jalview v.2.11.1.4 [41], and the tree obtained using PPlacer was exported to the Newick format to be visualised and manipulated in iTOL v.6 [42].

Taxonomic labels of OTU representative sequences were retrieved from the BLASTX best hit versus both McrA databases described (UniProt and Luke’s), and the NCBI taxonomy was retrieved from a taxIDs list using TaxonKit v.0.8.0 [43].

### 2.6. Whole Metagenome Shotgun Data

Metagenomic shotgun sequences from Area 5 were kindly provided by Brad Bebout from the NASA Ames Research Center. These metagenomic data were also previously used to describe the diversity of fungi in this hypersaline microbial mat [44]. Metagenomic sequence data are available through NCBI at BioProject PRJNA688760. For the exploration of methanogenic archaea presented in this study, quality control of raw reads was performed using FASTP v.1.14.5 with the default options. Metagenomic contigs were used for the annotation of methane metabolic pathways with the DRAM v.1.2.4 software [45] and the eggNOG-mapper v.2.1.3 [46]. An additional screening on the raw reads was performed to explore the presence of particular functions, such as methylotrophic methanogenesis and hydrogen-dependent methylotrophy, using the Short-Pair.py v.1.0 [47] software.

## 3. Results

### 3.1. Environmental Variables and Microbial Diversity

All physicochemical properties measured in this study are shown in Table 1. The lowest salinity of 60.6‰ corresponded to A1, whereas the highest salinity was 123.6‰ for A5. The temperature ranged from 29.5 °C in A1 to 24.7 °C in A5, and the dissolved oxygen concentration ranged from 6.2 to 8.0 mg/L. A total of 203,624 raw reads were obtained from the 16S rRNA gene sequencing. After denoising and chimera verification, 162,167 high-quality sequences were retrieved. Data were normalised by subsampling to the lowest read count (10,700); of this subset of sequences, 124,239 reads were affiliated with bacteria and 3402 with archaea. All denoising statistics are summarised in Appendix A. Shannon index values varied from 9.3 to 12.9 (Table 1), while the observed bacterial ASVs were in an average range of 354–644 for bacteria and 5–61 for archaea (Table 1). The bioinformatic analysis of the 16S rRNA gene sequences exhibited microbial communities that were dominated by bacteria, whereas archaeal members were detected in low relative abundances (0.2–4.4%) (Appendix A). This information was also supported by the metagenomic analysis (shotgun) performed on the A5 sample (Appendix A). 

The community composition analyses of the retrieved 16S rRNA ASVs for the bacteria domain showed that the microbial mats from A1 were mainly composed of Bacteroidia (40–37%), Alphaproteobacteria (6–9%) and Gammaproteobacteria (3–6%) and Spirochaetia (6–7%), while the dominant taxa from A4N1 belonged to Bacteroidia (39–43%), Spirochaetia (9–10%), Gammaproteobacteria (8%) and Alphaproteobacteria (6–8%). The mat samples from A4N5 were dominated by Bacteroidia (38–47%), Alphaproteobacteria (17–22%), Gammaproteobacteria (6–10%) and Cyanobacteria (3–8%). Finally, A5 displayed Bacteroidia (25–34%), Cyanobacteria (11–16%) and Spirochaetia (6–8%) as dominant members (Figure 1A).

For the archaeal domain, the phylogenetic assignment with the SILVA database showed Nanoarchaeota as the dominant phylum, followed by Thermoplasmatota, Asgardarchaeota and Euryarchaeota. Moreover, Nanoarchaeota exhibited the highest ASV richness in all four studied sites, although each area showed particular types of ASVs (Appendix A). At the taxonomic class level, Nanoarchaeia was the dominant group in all studied sites, followed by Thermoplasmata in A1, A4N1 and A5 (Figure 1B). In addition, a total of twelve different archaeal classes, belonging to eight different phyla, were detected in the studied sites (Figure 1B; Appendix A).

For the beta diversity analysis, the estimated PCoA on the weighted UniFrac distance matrix showed significant differences among the archaeal community structures for each site. A clear clustering pattern among the replicates of each site was observed. In general, the second principal coordinate explained that most of the variation was due to the A4N5 site (Figure 2).

### 3.2. Composition and Phylogeny of Methanogenic Archaea

Methanogenic archaeal members were evidenced using 16S rRNA sequencing, corresponding to the orders Methanosarcinales and Ca. Methanofastidiosales, and Methanobacteriales of the classes Methanosarcinia, Thermococci and Methanobacteria, respectively, all with relative abundances < 1% (Appendix A). The bioinformatic analysis of the *mcrA* sequences evidenced the presence of methanogenic members of the orders Methanomicrobiales, Methanosarcinales and Methanomassiliicoccales in the studied sites (Appendix A). Specifically, for sample A5, the shotgun annotation based on k-mers indicated the presence of Methanosarcinales, Methanomicrobiales, Methanocellales, Methanobacteriales, Methanococcales, Methanopyrales and Methanomassiliicoccales (Appendix A).

Due to the lack of available environmental sequences in the *mcrA* gene databases, the assignment of the sequences obtained at the lower taxonomic levels was limited. Thus, a phylogenetic tree was reconstructed to gain information about the relationship of the non-assigned *mcrA* sequences (Figure 3). The phylogenetic tree allowed the recognition of sequences closely related to clades in the Methanomicrobiales order, together with reported sequences of the genera *Methanolacinia*, *Methanosphaerula* and *Methanoregula*, with the latter being the most abundant (*n* = 198). For the Methanosarcinales and Methanotrichales orders, sequences were clustered with *Methanosarcina*, *Methanohalobium*, *Methanococcoides*, *Methanothrix*, *Methanolobus*/*Methanomethylovorans* and *Methanohalophilus*. In general, the sequences related to methanogenic archaea in the order Methanosarcinales, genera *Methanohalophilus* and *Methanolobus*/*Methanomethylovorans*, were less abundant, although *Methanohalophilus* showed a higher abundance (*n* = 69) in comparison with *Methanosphaerula* (*n* = 64) (Figure 3). In addition, a few sequences were assigned to the Methanomassiliicoccales cluster (Figure 3). Remarkably, two clusters of sequences retrieved in this study were not related to any other known methanogenic group and were, therefore, considered environmental methanogenic clusters presumptively specific to Guerrero Negro (Figure 3). To expand the knowledge of these clusters, additional experimentation and analysis related to their methanogenic metabolism will be necessary in order to assign them to accurate phylogenetic affiliations.

### 3.3. Insights into the Metabolic Pathway of Methanogens

Based on the detection of methanogenic members of the order Methanomicrobiales and Methanosarcinales, we hypothesised that hydrogenotrophic, acetoclastic and methylotrophic methanogenic metabolic pathways could be present in the hypersaline microbial mats from Guerrero Negro. This information was further explored using a shotgun metagenomic analysis of the A5 sample, in which several genes related to the three metabolic pathways were detected (Table 2).

To assess the relevance of the present genes to the three metabolic pathways for methane generation, the significant matches and relative abundances of these genes were estimated (Table 2). The annotation process revealed genes encoding methyl-compound methyltransferases (*mta*, *mtm*, *mtb*, *mtt*), reinforcing that the methylotrophic pathway is one of the best represented in hypersaline environments. In addition, the genes associated with autotrophic hydrogenotrophic methanogenesis (*fwd*, *ftr*, *mch*, *mtd*, *mer*, *mtr*), that encode the conserved core enzymes of this pathway, as well as the genes (*ack*, *acs*, *cdh*) encoding the key enzymes that use acetate for methane production, were also detected.

## 4. Discussion

Extreme environments are usually considered ecosystems with reduced biological complexity due to extreme salinity, temperature and solar radiation [48]. Accordingly, the microbial mats from Guerrero Negro, Baja California Sur, Mexico, have been considered simple systems, dominated by cyanobacteria and sulphate-reducing bacteria [49]. However, recent studies [50,51] have reported that the microbial communities of ESSA at Guerrero Negro are highly complex, with an unexpected diversity. The results obtained in this study allow for an increased knowledge of the unexplored archaeal diversity in these hypersaline microbial mats (Figure 1B).

The archaeal 16S rRNA analysis showed that six of the ten recovered phyla were assigned to the recently proposed and debated super-cluster DPANN, with a remarkable dominance of the phylum Nanoarchaeota in all samples analysed (Appendix A). This phylum, as well as several others of the under-studied members of DPANN, are distinguished by reduced cell sizes, genes and genomes, rapidly evolving gene sequences and the absence of some primary biosynthetic core genes, such as those involved in respiration and ATP synthesis, which confer them with limited metabolic capacities [52,53,54]. These genetic characteristics translate into an evolved dependence as a mutualist, commensalistic or parasitic, ecto-endo-symbiotic lifestyle [53,55]. In addition, their presence also evidenced that the microbial mats from Guerrero Negro harbour symbiotic or syntrophic lifestyles and highly diverse archaeal populations with a high number of uncultivated low-abundance species with unknown physiologies.

Several phyla in the DPANN super-cluster have shown a symbiotic lifestyle, which could explain why a particular ASV signature in the order Micrarchaeales appeared in high relative abundance, together with some signatures of members assigned to the class Thermoplasmata in site A5. Particular species of these taxa, as well as members of the Nanoarchaeota, have been reported to physically interact through pili-like structures [52,56]. Furthermore, the shotgun analysis of A5 also showed the presence of both types of symbiont members, specifically the presence of Ca. Microarchaeum sp. and Ca. Mancarchaeum acidiphilum of Micrarchaeota, as well as *Cuniculiplasma divulgatum* of Thermoplasmata (Appendix A). It is important to highlight that most of the interactions discovered so far occurred between acidophilic or thermophilic members of both taxa [55,57]. Therefore, the recovery of relatively strong signals of these archaea in a hypersaline environment (12.36%), and with a moderately alkaline pH (8.34), suggested that members of these taxa could have a wider range of tolerance to different physicochemical conditions [55,58], and, hence, play potentially different ecological roles in this environment.

A previous study [59] found a strong co-occurrence between Woesearchaeotales/Woesearchaeales and methanogens (Methanomicrobia and Methanobacteria), proposing a syntrophic metabolic model by a consortium of H_2_/CO_2_-using and acetate-using methanogens and members of the order Woesearchaeles, phylum Nanoarchaeota/phylum Ca. Woesearchaeota. This hypothesis could explain the observed high abundances of both Nanoarchaeota and Methanomicrobiales in our study, opening new perspectives regarding the possible interactions among them.

In turn, our results exhibited a low abundance of the recently discovered archaea in the new class Lokiarchaeia (superphylum Asgard). This group has been considered as one of the major achievements regarding the exploration of uncultivated diversity in this domain [60,61]. Although all four sites had the same dominant phyla in their archaeal communities, the estimated PCoA in the weighted UniFrac distance matrix showed significant differences in the community structure among the sites (Figure 2). These differences were consistent despite the observation of a small interaction of ASVs among sites, indicating the presence of site-specific phylotypes (Supplementary Material File S1). The results suggested that highly diverse populations with low–abundant ASVs (rare biosphere) were important in shaping the community structure and hence, represented a reservoir of genetic diversity that actively responded to environmental perturbations [62,63,64]. Moreover, the results of the beta diversity analysis pinpointed that the archaeal populations were adapted to the specific environmental conditions at each site, specifically salinity, which was different at each site (Table 1).

### 4.1. Methanogenic Diversity in Hypersaline Microbial Mats

The taxonomic assignment based on the 16S rRNA and *mcrA* sequences, and metagenomic analysis, evidenced the presence of Methanosarcinales, which was the better-studied methanogenic order in hypersaline environments [5,6,8,65]. However, this study also evidenced the presence of the orders Methanobacteriales, Methanomicrobiales, Methanomassiliicoccales, Ca. Methanofastidiosales, Methanocellales, Methanococcales and Methanopyrales, although some of these were present at extremely low relative abundances (Appendix A, Appendix A), suggesting that the hypersaline microbial mats from Guerrero Negro harboured a previously unexplored diversity of methanogens. It should be noted that the results might be limited by the reference databases used for the taxonomic classification of each gene. These findings highlight the necessity of using a variety of marker genes to characterise a broader spectrum of the methanogenic diversity residing in this ecosystem. Similar results have been reported for anaerobic digesters [66], but studies in extreme environments are scarce [67].

The lack of new cultures of methanogens [64] has limited the assignment of the environmental *mcrA* sequences at low taxonomic levels. Several *mcrA* environmental sequences obtained in this study were significantly different from those previously reported and did not match with any known methanogenic archaea, suggesting the presence of specific environmental clusters of methanogenic archaea in Guerrero Negro. The recovery of the genomes of uncultured groups from environmental metagenomes could result in the description of several new higher taxonomic levels, such as phyla [4]. However, since only a small fraction of incomplete archaeal genomes was assembled in this study due to their low abundance, further deeper sequencing or additional strategies to recover archaeal genomes from metagenomes are needed.

The strains recovered from hypersaline lakes have resulted in a new class of archaea (Halobacteria) that are methyl-reducing methanogens that use C1 methylated compounds as electron acceptors, and H_2_ or formate as electron donors [68]. We presumed that the unknown environmental Guerrero Negro clusters, closely related to the Methanonatronarchaeia archaeon and unrelated to the hydrogenotrophic, acetoclastic and methylotrophic methanogen members, could be relatives of this new class of Euryarchaeota (Figure 3). We observed the presence of sequences belonging to the class Thermoplasmata in all sampled sites (Figure 1B), which could be related to uncultured archaeal lineages. This group of uncultured archaea is poorly studied, and new members have been described as hydrogen-dependent methylotrophs.

No signatures belonging to the methanogenic members of the groups Verstraetearchaeota, Bathyarchaeota, Hadesarchaeota or Nezhaarchaeota, were detected. Additionally, no sequences from methanotrophic groups (i.e., ANME Class Methanomicrobia, Helarchaeota or Korarchaeota) were identified. These results suggested that methanogenesis in hypersaline microbial mats were restricted to Euryarchaeota and Thermoplasmata, whereas anaerobic methane oxidation was absent [69]. However, methane oxidation by aerobic members of bacteria cannot be ruled out. 

Until recently, all known species of aerobic methanotrophs belonged to the phylum Proteobacteria, in the classes Gammaproteobacteria and Alphaproteobacteria. However, thermoacidophilic methanotrophs were described that represented a distinct lineage within the bacterial phylum Verrucomicrobia. In this study, sequences with low relative abundances, assigned to the family Methylacidiphilaceae, phylum Verrucomicrobia, were detected. These were described in the literature as obligate aerobic methylotrophs, capable of growth on methane and methanol. Thus, we hypothesise that these groups can consume the methane produced by methanogenic archaea in the studied sites. Draft genomic analyses showed that the pmoCAB operon structure was the same as observed in proteobacterial methanotrophs, and phylogenetic analyses demonstrated that the proteobacterial and verrucomicrobial pmoA genes that encode to particulate methane monooxygenase evolved from a common ancestor [70]. In addition, 16S rRNA sequences were detected for the phylum Gemmatimonadota, which were potentially capable of aerobic methanotrophy, as was indicated by the detection of genes that encode methane monooxygenase, *pmoA*, *mmoA* [71].

### 4.2. Methanogenic Metabolism in Hypersaline Microbial Mats

Methylotrophic metabolism has been considered the only methanogenic pathway occurring within hypersaline environments [72,73]. Accordingly, we found the presence of several methylotrophic members of the order Methanosarcinales, as well as methylotrophic genes, in the metagenome. Furthermore, putative hydrogenotrophic members related to Methanomicrobiales have also been reported for this environment [8]. In this study, we hypothesised that all four of the methanogenic pathways could occur in hypersaline environments, based on functional inference and the detection of specific genes in the metagenome. Methylotrophic and acetoclastic methanogens in the Methanosarcinales order were well represented (Appendix A). Moreover, hydrogenotrophic methanogenic members of the orders Methanobacteriales, Methanomicrobiales, Methanocellales, Methanococcales and Methanopyrales were also observed (Appendix A). The presence of hydrogen-dependent methylotrophic methanogens was also evidenced by the detection of members related to the orders Methanomassiliicoccales and Ca. Methanofastidiosales (Appendix A), and by the detection of genes in the metagenome that were associated with methylotrophic metabolism, presumably lacking the Wood–Ljungdahl pathway [15].

Although the first isolates and cultures of Methanomassiliicoccales were obtained from human faeces, termite guts and water treatment sludge [74,75,76], recently, it was reported that metagenome-assembled genomes (MAGs) in this clade were recovered from natural environments [77,78,79,80]. Through the 16S rRNA and *mcrA* amplicon sequencing approaches used in the current study, the members that belong to the methanogenic taxon Ca. Methanofastidiosa and presumptive members closely related to the order Methanomassiliicoccales were recovered from our samples. To our knowledge, this is the first report describing members of microbial communities harbouring hydrogen-dependent methylotrophic methanogenesis metabolism recovered from a hypersaline environment. These findings provide insight regarding their ecological importance and suggest that Methanofastidiosa and Methanomassiliicoccales can thrive in different environments and that high salinity does not limit their presence.

The bacterial composition analysis (Appendix A) showed that the specific bacterial phyla recovered had key roles as detritus and polysaccharide degraders, and were also reported to produce extracellular proteases, glycosyl hydrolases and lipases [81,82,83,84,85]. Additional members of these groups and other bacterial groups were also retrieved. Together, these groups were described to be involved in the subsequent steps of organic matter degradation, through the fermentative, acidogenic and acetogenic pathways, which ultimately gave rise to methane formation [86,87].

In hypersaline environments, methylated compounds play a central role in the production of methane [65,73] and their occurrence may be explained through the conversion of osmolytes, such as glycine-betaine and choline up to trimethylamine, by representatives of the classes Clostridia [88], Halanaerobiia [89], and by sulphate-reducing bacteria [90]. Moreover, the trimethylamine N-oxide (TMAO) can be reduced to TMA by the bacterial genera *Alteromonas*, *Flavobacterium*, and halophilic archaea [88,91]. This could be consistent with the results described by [92], who pointed out the requirement of H_2_/methyl substrates and acetate for the growth and methanogenic activity of Methanomassiliicoccales. Furthermore, the apparent adaptation of Methanomassiliicoccales to thrive in sediments with high sulphate concentrations [93], as well as the specialisation of Ca. Methanofastidiosa in the use of methylated thiols as a methanogenic substrate [94], not only establishes a bridge between the carbon and sulphur cycles in eutrophic environments but also potentially contributes to the regulation of the H_2_ partial pressure in the microbial mats.

On the other hand, given that hypersaline environments exhibit a high rate of sulphate reduction [95], the finding of genes associated with all methanogenic pathways (Table 2) suggests the coexistence of not only novel methanogens, but also hydrogenotrophic and acetoclastic groups with sulphate-reducing bacteria, despite their competition for H_2_ and acetate [96]. Similar coexistence patterns were observed in non-oligotrophic environments, such as estuarine and marine sediments, tropical coastal lagoons and mangroves [77,80,97,98]. These results support the hypotheses of the occurrence of putative hydrogenotrophic methanogens in the ecological functioning of hypersaline microbial mats.

## 5. Conclusions

The different genomic and metagenomic approaches presented in this study allowed a comprehensive exploration of four hypersaline environments with an interesting component of highly diverse, extremely low-abundance archaea, with unknown physiologies. We identified different metabolic types of previously undetected methanogens, hydrogenotrophs, acetoclastic and hydrogen-dependent methylotrophs. Our results provided evidence for the coexistence of all methanogenic metabolism pathways, rather than the exclusive presence of methylotrophic methanogenic archaea in hypersaline environments, as previously considered [99,100]. The integration of the obtained results through the bioinformatic analyses revealed that the decomposition of organic matter with concomitant methane production was the consequence of complex trophic interactions between microbial guilds that involved interspecies hydrogen transfer between organotrophs and methanogens. These findings shed light on the hitherto disregarded participation of bacteria in the methanogenic carbon cycle, and we are providing new insights into anaerobic microbial ecology in hypersaline microbial mats.

## Figures and Tables

**Figure 1 microorganisms-11-00812-f001:**
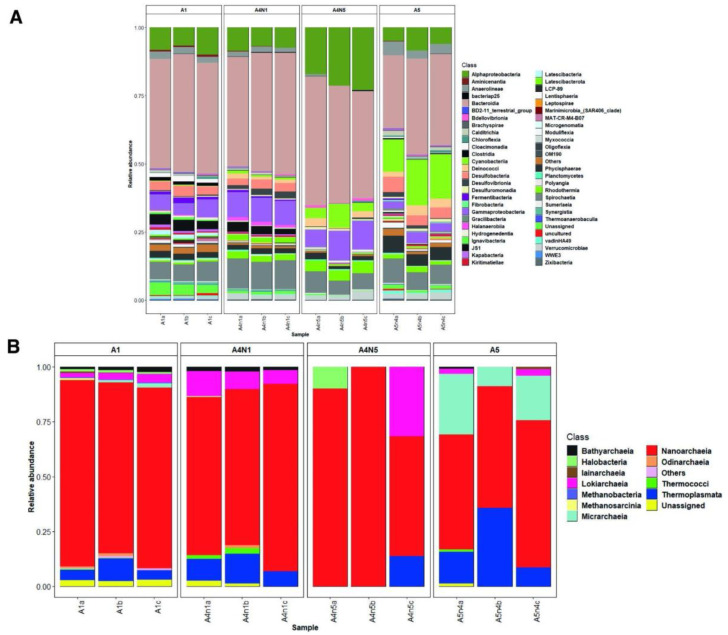
Stacked bar plots showing the relative abundances of bacterial (**A**) and archaeal (**B**) populations across all microbial mat samples at the class level. Taxa representing < 0.1% are grouped in “others”.

**Figure 2 microorganisms-11-00812-f002:**
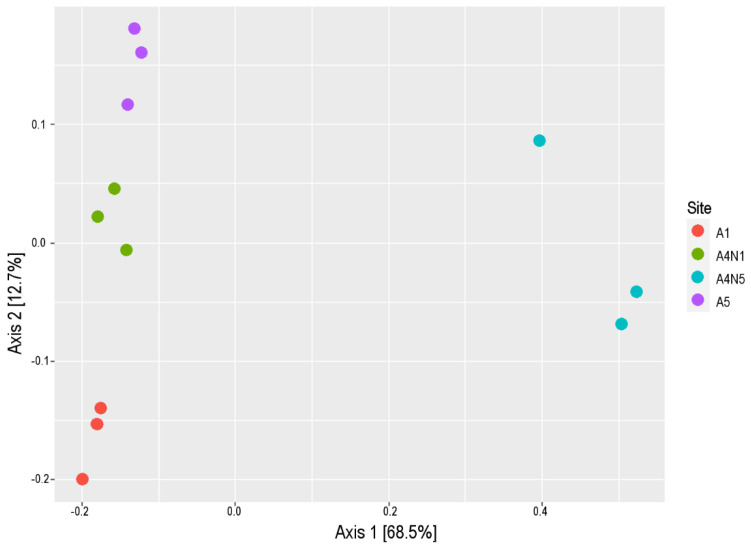
Principal Coordinate Analysis based on weighted UniFrac, estimated on archaeal 16S rRNA gene amplicon sequences from microbial mat samples.

**Figure 3 microorganisms-11-00812-f003:**
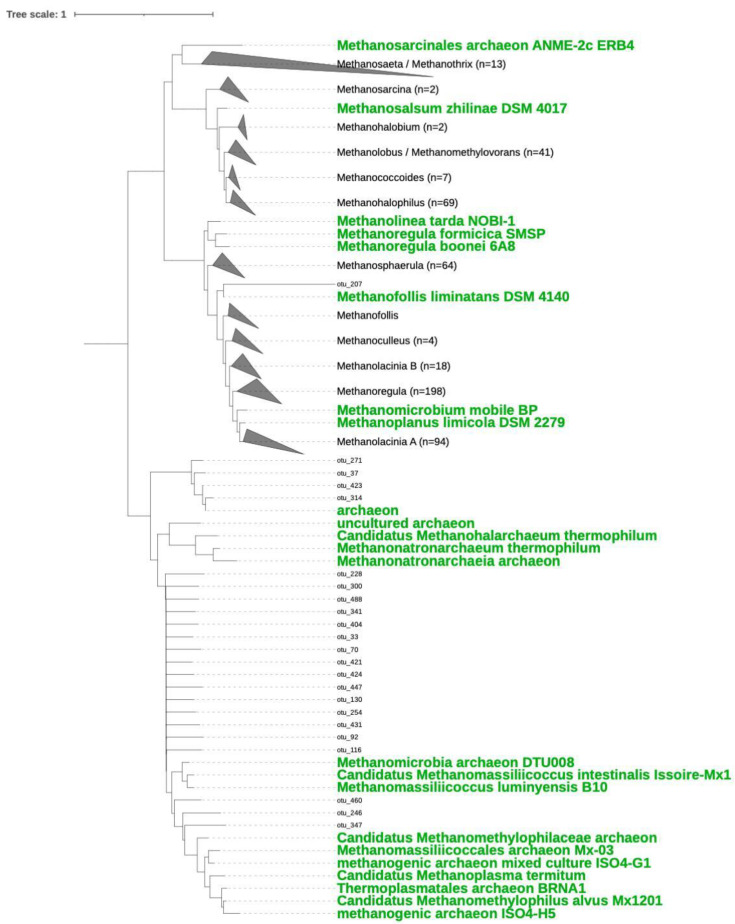
Phylogenetic tree of representative OTUs representative of McrA amino acid sequences in relation to a reference tree. Collapsed groups are represented with triangles, and the clade is labelled after the closest reference. The numbers of OTUs in the collapsed branches are shown in brackets. Terminal nodes corresponding to referenced sequences are labelled in green.

**Table 1 microorganisms-11-00812-t001:** Sampling sites, physicochemical parameters at Exportadora de Sal, S.A., Guerrero Negro, BCS, México and calculated alpha diversity from 16S rRNA archaeal sequences. Values corresponding to the arithmetic mean of triplicate measurements of the interstitial water from microbial mat samples. Dissolved oxygen (D.O.) was measured in the laboratory without replicates. Abbreviation codes for each sample are bracketed in parentheses.

Sample	Coordinates	Salinity (‰)	Temp. (°C)	D.O. (mg/L)	pH	Average Bacterial Observed ASV	Average Archaeal Observed ASV	Shannon Index
Area 1 (A1)	27.364 N, 113.539 W	60.6 ± 4.72	29.5 ± 0.40	7.2	8.52 ± 0.005	644 ± 16	61 ± 14	12.9 ± 0.1
Area 4 Near Area 1 (A4N1)	27.601 N, 113.8969 W	83.3 ± 4.04	27.6 ± 0.81	6.2	8.62 ± 0.005	471 ± 25	26 ± 5	11.1 ± 0.2
Area 4 Near Area 5 (A4N5)	27.690 N, 113.9210 W	118.3 ± 2.88	26.2 ± 0.46	8.0	8.32 ± 0.005	354 ± 51	5 ± 4	9.3 ± 0.8
Area 5 (A5)	27.690 N, 113.9209 W	123.6 ± 0.57	24.7 ± 0.17	7.0	8.34 ± 0.005	413 ± 71	36 ± 9	10.5 ± 0.8

**Table 2 microorganisms-11-00812-t002:** Relevant genes related to the three metabolic pathways for methane generation, detected through the metagenomic analysis of the A5 sample. The right column corresponds to significant matches using the raw metagenomic reads (96,506,197 of paired reads tested).

Gene	E.C. Number	PFAM	Module	Matching Paired Reads	Proportion (%)
*cdhC*	EC:2.3.1.169	PF03598	acetate => methane	1585	0.0016
*pta*	EC:2.3.1.8	PF01515	acetate => methane	9999	0.0104
*ackA*	EC:2.7.2.1	PF00871	acetate => methane	10,586	0.0110
*acs*	EC:6.2.1.1	PF16177	acetate => methane	2523	0.0026
*mtrA-H*	EC:2.1.1.86	PF04208	acetate => methane	362	0.0004
			CO2 => methane		
*mer*	EC:1.5.98.2	PF00296	CO2 => methane	3179	0.0033
*mtd*	EC:1.5.98.1	PF01993	CO2 => methane	69	7 × 10^−5^
*hmd*	EC:1.12.98.2	PF03201	CO2 => methane	83	9 × 10^−5^
*mch*	EC:3.5.4.27	PF02289	CO2 => methane	864	0.0009
*ftr*	EC:2.3.1.101	PF01913	CO2 => methane	698	0.0007
		PF02741	CO2 => methane	620	0.0006
*fwdA, fmdA*	EC:1.2.7.12	PF01493	CO2 => methane	4823	0.0050
		PF01568		4732	0.0049
*mtaA*	EC:2.1.1.246	PF01208	methanol => methane	18,823	0.0195
*mtaB*	EC:2.1.1.90	PF12176	methanol => methane	1449	0.0015
*mtbA*	EC:2.1.1.247	PF01208	methylamine/dimethylamine/trimethylamine => methane	18,823	0.0195
*mttB*	EC:2.1.1.250	PF06253	methylamine/dimethylamine/trimethylamine => methane	47,197	0.0489
*mtbB*	EC:2.1.1.249	PF09505	methylamine/dimethylamine/trimethylamine => methane	653	0.0007
*Dmd*	EC:1.5.8.1	PF00724	methylamine/dimethylamine/trimethylamine => methane	7091	0.0073
*Tmd*	EC:1.5.8.2	PF07992	methylamine/dimethylamine/trimethylamine => methane	71,733	0.0743
*mtmB*	EC:2.1.1.248	PF05369	methylamine/dimethylamine/trimethylamine => methane	1750	0.0018
*hdrA1*	EC:1.8.7.3	PF00037	methylamine/dimethylamine/trimethylamine => methane	56,679	0.0587
		PF02662	methanol => methane	7902	0.0082
		PF07992	acetate => methane	71,733	0.0743
*hdrABC*	EC:1.8.98.4	PF00037	methylamine/dimethylamine/trimethylamine => methane	56,679	0.0587
	EC:1.8.98.6	PF12838	acetate => methane	35,515	0.0368
	EC:1.8.98.5	PF07992	CO2 => methane	71,733	0.0743
*hdrD*	EC:1.8.98.1	PF02754	methylamine/dimethylamine/trimethylamine => methane	10,925	0.0113
		PF13183	methanol => methane	13,950	0.0145
			acetate => methane		
			CO2 => methane		
*mcrA*	EC:2.8.4.1	PF02249	methylamine/dimethylamine/trimethylamine => methane	139	0.0001
		PF02745	methanol => methane	188	0.0002
		PF02241	acetate => methane	166	0.0002
		PF02783	CO2 => methane	91	9 × 10^−5^
		PF02240		151	0.0002
*Fhs*	EC:6.3.4.3	PF01268	C1-unit interconversion; Wood–Ljungdahl pathway	16,065	0.0166
		PF00763	C1-unit interconversion; Wood–Ljungdahl pathway	3503	0.0036
		PF02882	C1-unit interconversion; Wood–Ljungdahl pathway	8293	0.0086
*Fdh*	EC:1.17.1.9	PF04879	Wood–Ljungdahl pathway	4417	0.0046
		PF00384	Wood–Ljungdahl pathway	12,428	0.0129
		PF01568	Wood–Ljungdahl pathway	4732	0.0049
*cdhA*	EC:1.2.7.4	PF03063	CO2 => acetyl-CoA; Wood–Ljungdahl pathway	21,268	0.0220

## Data Availability

The datasets generated during and/or analysed during the current study are available in the NCBI database under the BioProject accession number PRJNA821506.

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
