# Peer review of "Revisiting Microbial Diversity in Hypersaline Microbial Mats from Guerrero Negro for a Better Understanding of Methanogenic Archaeal Communities"

_microorganisms, 2023, doi:10.3390/microorganisms11030812_

Round 1

Reviewer 1 Report

In this manuscript the authors aims to analyze the diversity of methanogenic archaeal communities from Guerrero Negro, through amplicon sequencing of 16S rRNA and mcrA genes, and also making a comparison with shotgun metagenome sequencing data, trying to evaluate the metabolic potential of methanogenic pathways in these hypersaline ecosystems.

The authors found and identified different metabolic types of previously undetected methanogens (hydrogenotrophs, acetoclastic and hydrogen-dependent methylotrophs).

This study provide evidence for the coexistence of diverse methanogenic metabolisms. Besides, shed light on the participation of bacteria in the methanogenic carbon cycle, contributing to the knowledge of this kind of extreme-ecosystems.

For improvement of this study, I would like propose few points: 

- I cannot find Table 1.

- As a suggestion, supplementary material may be rearrange in an easy way for the reader. Only one or 2 supplementary information files, with subtitles explaining tables and figures inside. I think the names of table S1 and table S2 are changed? See supplementary material titles

-The legends of table 1 and table 2 should be on the main text and not in the supplementary information. Besides I cannot find Table 2 in the main text or in supplementary…

-PCA should goes in the main text

Author Response

- I cannot find Table 1.

Answer: We have added Table 1 and its description into the manuscript (page 5, line 94-98).

- As a suggestion, supplementary material may be rearrange in an easy way for the reader. Only one or 2 supplementary information files, with subtitles explaining tables and figures inside.

Answer: We re-arranged the supplementary materials as requested. The supplementary figures and tables were embodied in a single document and its order agrees with the order of mentioning in the manuscript. In addition, captions for each supplementary figure and tables were added.

- I think the names of table S1 and table S2 are changed? See supplementary material titles

Answer: Supplementary table S1 and S2 names were corrected. Moreover, the Supplementary Excel files were corrected in the names of the tabs in the following way:

File "Supp_mat_S1_MDPI.xlsx", the name of the description tab change from "Table S2 Description" to "Supp Mat 1 Description"

File "Supp_mat_S2_MDPI.xlsx"; the name of the description tab change from "Table S3 Description" to "Supp Mat 2 Description"

- The legends of table 1 and table 2 should be on the main text and not in the supplementary information. Besides I cannot find Table 2 in the main text or in supplementary.

Answer: Legends of Table 1 and 2 were now included; page 5, line 94-98 and page 16, line 295-296, respectively.

- PCoA should goes in the main text.

Answer: Attending the suggestion of the reviewer we have added the previous "Supplementary figure S3 - PCoA chart", as a new Figure 2 embodied into the manuscript. The new figure and its description is now located at page 12, lines 238-240.

Reviewer 2 Report

The authors present an intriguing paper dealing with methanogenic archaeal communities in hypersaline mats from Guerrero Negro. Methanogens in such an environment are pretty difficult to investigate especially due to problems to cultivate and low abundance. The authors were successful based on 16s rRNA, metagenomic and mcrA analyses to find taxa of the Methanosarcinales and from other related taxa in extremely low abundances. They conclude that all four known methanogenic pathways occur in this lake. The presence of these metabolic pathways in such an environment is a new milestone in archaeal research. The provided methods are top state of the art and could not find any mistakes or misinterpretations. Six of the ten recovered phyla were assigned to the super-cluster DPANN, with a dominance to the phylum Nanoarchaeota. For me also important is the knowledge that these forms harbor probably a strong symbiotic lifestyle. Also the observation that methanotrophic groups like ANME are missing and only methanogenesis is restricted to members of the Euryarchaeota and Thermoplasmata. Figures are well made and also the paper is well structured – this also valid for the supplementary informations. In any case this paper is a highlight in the understanding of extreme saline environments and give inspirations to look for these microbial communities also in more extreme environments.

Author Response

Thank you so much for the positive comments.